# Role of Survivin in Bladder Cancer: Issues to Be Overcome When Designing an Efficient Dual Nano-Therapy

**DOI:** 10.3390/pharmaceutics13111959

**Published:** 2021-11-19

**Authors:** Maria Arista-Romero, Anna Cascante, Cristina Fornaguera, Salvador Borrós

**Affiliations:** 1Grup d’Enginyeria de Materials (Gemat), Institut Químic de Sarrià (IQS), Universitat Ramon Llull (URL), Via Augusta 390, 08017 Barcelona, Spain; marista@ibecbarcelona.eu (M.A.-R.); anna.cascante@iqs.url.edu (A.C.); cristina.fornaguera@iqs.url.edu (C.F.); 2Sagetis Biotech SL, Via Augusta 394, 08017 Barcelona, Spain

**Keywords:** bladder cancer, polymeric nanoparticles, combined nano-therapies, paclitaxel, survivin

## Abstract

Bladder cancer is the 10th most diagnosed cancer, with almost 10 M cancer deaths last year worldwide. Currently, chemotherapy is widely used as adjuvant therapy after surgical transurethral resection. Paclitaxel (PTX) is one of the most promising drugs, but cancer cells acquire resistance, causing failure of this treatment and increasing the recurrence of the disease. This poor chemotherapeutic response has been associated with the overexpression of the protein survivin. In this work, we present a novel dual nano-treatment for bladder cancer based on the hypothesis that the inhibition of survivin in cancer cells, using a siRNA gene therapy strategy, could decrease their resistance to PTX. For this purpose, two different polymeric nanoparticles were developed to encapsulate PTX and survivin siRNA independently. PTX nanoparticles showed sizes around 150 nm, with a paclitaxel loading of around 1.5%, that produced sustained tumor cell death. In parallel, siRNA nanoparticles, with similar sizes and loading efficiency of around 100%, achieved the oligonucleotide transfection and knocking down of survivin expression that also resulted in tumor cell death. However, dual treatment did not show the synergistic effect expected. The root cause of this issue was found to be the cell cycle arrest produced by nuclear survivin silencing, which is incompatible with PTX action. Therefore, we concluded that although the vastly reported role of survivin in bladder cancer, its silencing does not sensitize cells to currently applied chemotherapies.

## 1. Introduction

Bladder cancer is the 10th most diagnosed cancer with an estimated 440,000 new cases worldwide in 2020, accounting for a 5-year prevalence of up to 1.7 M people [1,2]. The most common histologic type of bladder cancer is urothelial carcinoma, which was formerly known as transitional cell carcinoma (TCC), due to the fact that the transitional cells located at the outside layer of the bladder are the ones transformed into cancer cells [3]. Although current diagnosis methods are highly invasive, and bladder cancer remains complex and difficult to identify [4,5], it is well known that all types of bladder cancer start in the inner lining of the bladder but from different cell types: urothelial cells (95% of bladder cancer), squamous cells (4%) and glandular mucus cells (1%). Approximately 75% of patients are non-muscle-invasive TCC and have a 5-year survival rate between 88% and 98% [2,6]. The other 25% of patients diagnosed with TCC are muscle-invasive in stages between 1 and 4. Depending on the stage of the muscle-invasive TCC the survival rate can range between 80% in the 5 years after diagnosis until 5% of survival with cancer in stage 4 [1].

The standard of care first-line treatment for muscle-invasive TCC is radical cystectomy (transurethral resection) with adjuvant chemoradiotherapy (i.e., platinum-based as the most used, although not applicable to all patients due to extreme toxicities; paclitaxel as alternative) [2,7]. For unresectable, advanced-stage or metastatic urothelial carcinomas, the use of immune checkpoint inhibitors is recommended as second-line treatment, thanks to the increase in a patient’s tolerability. Although poor understanding of the molecular mechanisms involved in this type of cancer, gene therapies targeted to alterations involved in tumor growth are starting to gain a role after the description of some genes involved in the chemoresistance found in many patients, and they are being tested in multiple trials [2,6]. Thus, this scenario makes clear the need for a combined therapy to target, simultaneously, different but complementary mechanisms of bladder cancer tumor cells and achieve their selective death avoiding tumor cells’ drug resistance [8].

An interesting combination is the gene expression modulation to sensitize cells followed by chemotherapy, as performed for other cancer types [9,10], but if administered naked, both therapies could produce severe side effects that could hamper patients’ survival. In this context, the vehiculation of the drugs using nanomedicine tools, together with local delivery, will benefit patients’ tolerance while reducing side effects. Polymeric nanoparticles, after proper design, can accomplish both objectives. Firstly, they can protect the active ingredient and direct it to the target organ, the tumor, by the enhanced permeability and retention effect. Secondly, the concentration of the drug in the tumor will, in addition to decreasing side effects, enable a low dose administration, which will decrease therapy costs. Thirdly, they enable a sustained release of the drug, which decreases repeated doses, and fourthly, they enable the in vivo administration of nucleic acids, which, otherwise, will be degraded when in contact with physiological fluids [11,12,13,14].

Because the bladder is an easily accessible organ through intravesical administration, here we propose a local delivery of the particles, previously stated advantageous to decrease the invasiveness of conventional intravenous tumor therapies, as well as to avoid off-target effects of the therapy by increasing the exposure of the affected bladder lining to the drug [10,15]. Although previous studies remarked the difficult reproducibility of treatments efficacy when using this route, in our case, the use of nanosystems will facilitate the penetration of the active ingredients through the bladder wall to the target cells, one of the main bottleneck steps reported [7].

Taking all this into account, we aim to design a combination therapy based on polymeric nanoparticles including a chemotherapeutic drug with silencing oligonucleotides targeting a resistance-associated gene. Regarding the chemotherapeutic drug, paclitaxel was selected as a taxane example of retained activity and improved tolerability, as compared to traditional MVAC (methotrexate, vinblastine, doxorubicin and cisplatin). As we previously described [16,17], it was encapsulated in our proprietary block co-polymer P nanoparticles (rigid hydrophobic polyester block + flexible hydrophilic PEG block). As for silencing therapy, we selected survivin as the target gene, a protein that plays an important role in the suppression of apoptosis and regulation of cell division [9,18,19,20,21]. The overexpression of survivin in cancer allows the cell to overcome cell cycle checkpoints, facilitating the aberrant progression of transformed cells through mitosis and blocking the caspases pathway in the cytoplasm, hence, avoiding the apoptosis of a defective cell. Here, we used our poly (beta-aminoester) proprietary polymers, previously demonstrated to efficiently encapsulate and protect a variety of nucleic acids [22,23,24], to nano-encapsulate a small interfering RNA (siRNA) codifying for survivin since siRNAs were described to potentially interfere with mRNA expression [25,26]. Finally, we set up a dual combination therapy for bladder cancer patients.

## 2. Materials and Methods

*Materials:* MTT, BCA, propidium iodide and PVDF membranes were acquired from Merck. PTX was obtained from Hunxol I Yunnan Hande Bio-tech co (Yunnan, China). Lipofectamine^®^ 2000 reagent was obtained from Invitrogen (ThermoFisher Scientific, Waltham, MA, USA). Actin primary mouse antibody and goat anti-rabbit IgG HRP were purchased from Abcam (ab3280) (ABCam, Cambridge, UK). Survivin polyclonal primary rabbit antibody was obtained from Novus Biologicals (NB500-201) (Bio-Techne, Minneapolis, MN, USA). antibody was purchased from Abcam (ab6721 and ab97046) (ABCam, Cambridge, UK). Goat anti-rabbit IgG conjugated Alexa 488 was purchased from ThermoFisher (ThermoFisher Scientific, Waltham, MA, USA). Protein Bromelain (PB), poly (beta aminoesters (pBAEs) and polymer P were synthesized by other group members, as previously detailed [16,22,23,27]. siRNA non-targeting pool was obtained from Dharmacon (D-001 206-13-05) (GE Healthcare, CO, USA) and siRNA-F AF 546 was obtained from Qiagen (Qiagen, Germany). siRNAs anti survivin were obtained from Sigma Aldrich and have the following sequences 1: sense 5′-GGACCACCGCAUCUCUACA-3′, antisense 5′-UGUAGAGAUGCGGUGGUCC-3′; 2: sense 5′-GAACUGGCCCUUCUUGGAG-3′, antisense 5′-CUCCAAGAAGGGCCAGUUC-3′.

*Cell lines:* RT4 cells (ATCC^®^ HTB-2™; human urinary bladder, transition to cell papilloma) and T24 cells (ATCC^®^ HTB-4™; human urinary bladder, grade 3 transition to cell carcinoma) were purchased from ATCC (Manassas, VA, USA). Cells were maintained at 37 °C in 5% CO_2_ atmosphere in complete McCoy’s 5A medium, containing 10% fetal bovine serum, 100 units/mL penicillin, 100 ug/mL streptomycin and 1.5 mM L-glutamine. Cells were passaged every 2–3 days at 1/10 dilution rate and grown in P100 plates (surface area is 75 cm^2^).

*Synthesis of P polymer nanoparticles encapsulating PTX:* Nanoparticles (named PTX-NP) were prepared according to a modified nanoprecipitation method published before [16]. Briefly, 20 mg of P polymer dissolved in 600 μL of acetone and 400 μL of PTX (1 mg/mL PTX in acetone) were mixed together, to form the diffusing phase. This phase was then added to dispersing phase, 20 mL of water, by means of a syringe controlled by a syringe pump (KD Scientific), positioned with the needle directly in the medium, under a magnetic stirring of 700 rpm and at room temperature and with a flux of 50 μL/min. The resulting NPs suspension was then centrifuged in Amicon centrifugal filters at 6000 rpm two times, first 20 min and then for another 30 min. The filtered NPs were resuspended with 1 mL of mQ H_2_O.

*Synthesis of siRNA pBAE nanoparticles:* Two different types of pBAE-NPs were synthesized: with and without protein bromelain (PB) coating. Additionally, two different nucleic acids were encapsulated: siRNA and pGFP as reporter genes. NPs encapsulating siRNA had a final concentration of siRNA of 0.03 mg/mL and the different polymer: siRNA ratios used were 25:1, 50:1, 100:1, 150:1, 200:1 and 300:1 (*w/w*). For pGFP-nanoparticles, the polymer: DNA ratio used was 50:1 (*w/w*) and the final concentration of plasmid was 0.06 mg/mL. Polymers used to prepare both types of NPs were of C32 type, and different combinations were used: C32-CR3, C32-CK3, C32-CR3:C32-CK3, C32-CK3:C32-CH3 and C32-CR3:C32-CH3, named as follows: R, K, RK, KH and RH, with the following protocol we previously described [23]. Briefly, siRNA NPs were prepared by mixing equal volumes of siRNA at 0.03 μg/μL with polymers at different concentrations, depending on the polymer: siRNA ratio, in NaAc buffer solution (25 mM, pH 5.5). When the encapsulated genetic material was pGFP, the procedure was the same but with a concentration of 0.06 μg/μL. Then, siRNA was added over polymer solution and was mixed by pipetting, followed by vortexing for 5 s and was incubated at room temperature for 10 (especially for pGFP) or 30 min. When the complexes had coating of PB, different dilutions of PB were prepared, as previously described [27]. To create a coating of PB, siRNA was diluted the same way previously described and 2 μL of R(100 μg/μL) was diluted in 48 μL of NaAc buffer solution (25 mM, pH 5.5) to obtain a final concentration of 6 μg/μL (the concentration was doubled compared to previously due to the fact that is half the volume). After mixing and vortexing for 5 s of the mixture of siRNA and polymer, 50 μL of PB with the corresponding concentration were added carefully, followed by vortexing for 5 s and were incubated at room temperature for 10 (specially for pGFP) or 30 min.

*Determination of nucleic acid encapsulation by electrophoretic mobility shift assays:* The capacity of NPs to encapsulate siRNA at different polymer ratios was studied with the electrophoretic mobility of polymer: siRNA complexes, which was measured on agarose gels (2.5% of agarose *w/v*) in Tris-Acetate-EDTA (TAE) buffer containing ethidium bromide. The electrophoresis mixture was added into the cubed and the gel was allowed to solidify for 20 min. Then the electrophoresis support was placed into the TAE 1× bath. Finally, samples were loaded and were run for 1 h at 80 V (Apelex PS 305, France). Finally, siRNA bands were visualized by UV irradiation.

*Determination of NPs size and polydispersity:* Particle size and surface charge measurements were determined by dynamic light scattering (DLS) at room temperature with a Zetasizer Nano ZS (Malvern Instruments Ltd., Worcestershire, UK, 4-mW laser) using a wavelength of 633 nm. Correlation functions were collected at a scattering angle of 173°, and particle sizes were calculated using the Malvern particle sizing software (DTS version 5.03). The value was recorded as the mean +/− standard deviation of three measurements and each measurement was determined from the average of 20 cycles in a disposable plastic cuvette. The size distribution was given by polydispersity index. The zeta potentials of complexes were determined from the electrophoretic mobility by means of the Smoluchowski approximation. The zeta potential of samples was determined in triplicate from the average of 10 cycles of an applied electric field. In this case, 1 mL of the previous complexes were added into zeta potential cuvette.

*PTX loading efficiency:* Freeze-dried NPs loaded with PTX were dissolved in acetonitrile and the amount of entrapped drug was detected by Ultra Performance Liquid Chromatography (UPLC) (Waters ACQUITY UPLC H-Class). A reverse-phase BEH C18 column (1.7 μm 2.1 × 50 mm) was used. The mobile phase consisted of a mixture of acetonitrile and water (60:40 *v/v*) and was delivered at a flow rate of 0.6 mL/min. PTX was quantified by UV detection (λ = 227 nm, Waters TUV detector). Drug content was expressed as drug content (D.C. % *w/w*); represented by Equation (1). For each sample, the mean value was recorded as the average of three measurements. The results were expressed as mean ± S.D for two replicates. Equation (1): Calculation of drug content of encapsulation.
(1)$$Drug\;Content\;\left(\%\frac{w}{w}\right)\;=\frac{\mathrm{Mass}\;\mathrm{of}\;\mathrm{drug}\;\mathrm{in}\;\mathrm{NPs}\;\times 100}{\mathrm{Mass}\;\mathrm{of}\;\mathrm{NPs}\;\mathrm{recovered}},$$

*In vitro cellular transfection of pBAE-NPs:* For immunofluorescence experiments, siRNA F AF546b was used. Cells were grown over a sterile cover slip (gelatine at 0.1% coating for 20 min) in a 12-well plate. Cells were seeded at 200,000 cells/well and incubated overnight to 80% confluence. Cells were washed with PBS 1× and siRNA complexes were added diluted in Mccoy’s minimum medium at a final concentration of 16 pmol of siRNA/well. Then, cells were incubated for 2 h at 37 °C in 5% CO_2_ atmosphere. All the transfections and controls were performed in triplicate. For flow cytometry experiments, the experiments were performed equally but scaled down to 96 well plates, and pGFP was used instead. For Western blot analysis, on the contrary, the experiment was scaled up to 6-well plates.

*Cytotoxicity analysis by MTT assay:* Performed as we reported previously [16,24].

*Fluorescent microscopy to determine nanoparticle uptake:* After desired time, cells were washed with PBS 1× and then formalin 10% was added during 20 min at RT. Afterward, cells were washed twice with 1000 μL of PBS 1× and 100 μL of Triton-X-100 0.1% was added in order to allow the permeabilization of the cells. After 30 min cells were washed again twice with PBS 1× and were incubated with DAPI 1:10,000 in PBS 1× for 5 min. Finally, cells were washed three more times with PBS 1× for 5 min. The covers were prepared with mounting medium and were ready to be seen under fluorescence light. Fluorescence was analyzed with the corresponding filter with the fluorescence Zeiss Axiovert 200 M microscope. ImageJ was used for the quantification of the fluorescent signals, according to recommended protocol [28]. In brief, relative quantification (CTCF values) was performed by normalizing the regions of interest of the transfected cells to the black regions as background.

*Survivin expression by Western blot analysis:* (1) Cell lysis by aspirating media and cells were washed with warm PBS 1×. Then, cells were scraped, collected on Eppendorf tubes and centrifuged at 1500 rpm for 2 min at 4 °C. The pellets were dissolved and incubated with lysis buffer (RIPA reagent 1× and 1:200 Protein inhibition cocktail) for 20 min on ice. Next, centrifugation of lysate at 10.000 rpm for 10 min was performed and supernatants were stored at −20 °C in aliquots of 20 μL. (2) Protein quantification by BCA, following distributor instructions. It was necessary 30 μg of total protein for survivin protein study. (3) SDS-PAGE Gel preparation and running. Running gels: 15% acrylamide. Stacking gels: 6.1 mL of mQH_2_O, 2.5 mL of solution C (0.5 M Tris-HCl), 1.3 mL of solution A, 100 μL of solution D, 10 μL of TEMED and 50 μL of solution G. The samples had added loading buffer and 25 μL of sample was loaded in the gel. Gels were bathed with electrophoresis buffer (7.5 g Tris-basic, 39 g Glycine, 2.5 SDS and 50 mL of mQH_2_O) and run at 150 V (constant). (4) Transfer of the proteins to a PVDF membrane using the XCell IITM Blot Module from Biorad. Pre-wetting of the PVDF membrane in 100% methanol for 30 s, drain and equilibrate with transfer buffer (3.03 g Tris-basic, 14.4 g glycine, 200 mL methanol). The transfer run for 2 h at 40 V imbibed in transfer buffer. (5) Blocking and detection (actin + surviving). After the transfer, the membranes were incubated at room temperature for 2 h in an orbital shaker with blocking buffer (PBS 1×, 0.1% Tween and 5% non-fat powdered milk). Primary antibodies were resuspended in blocking buffer (Mouse anti-actin 1:2000; goat anti survivin 1:1000) and then were incubated with the membrane overnight at 4 °C in an orbital shaker. Next, the membranes were washed out with washing buffer three times for 10 min. The secondary antibody was resuspended in PBST (PBS 0.1% (*v/v*) Tween 20) (Goat anti-rabbit HRP 1:2000; Rabbit anti-mouse HRP 1:10,000) and it was incubated with the membrane. Next, the membrane was washed 3 times with PBST for 10 min, and HRP was detected by chemiluminescence with Luminata^TM^ forte. Then, the membrane was revealed using ImageQuant LAS 4000 mini (GE Healthcare Life Science).

*Survivin intracellular localization by immunofluorescence:* After the same treatment explained before for cell uptake, incubation with the primary antibody (dilution 1:100) previously described against survivin was made. The secondary antibody was goat anti-rabbit Alexa 488 at a dilution of 1:1000 A final washing step was performed with PBS 1× and DAPI staining was carried out as previously described. The mounting was made with mounting solution and the samples were studied under Zeiss microscope.

*Cell cycle analysis by flow cytometry:* Cell media after transfection were aspirated and cells were washed with warm PBS 1×. Then, cells were trypsinized and collected in Eppendorf tubes and centrifuged at 1000 rpm for 5 min. The pellet was washed with PBS 1×. Cells were centrifuged again at 1000 rpm for 5 min and pellet was resuspended with a solution of 70% of cold ethanol. For propidium iodide staining cells were centrifuged at 1000 rpm for 5 min and the ethanol was decanted. Cells were washed with PBS 1× and centrifuged again at 1000 rpm for 5 min. A mixture of 0.1% (*v/v*) Triton X-100 (Sigma) in PBS with 2 mg of RNasa A and 200 μL of propidium iodide 1 mg/mL was prepared. Cells were resuspended with this mixture at a concentration of 1 × 10^6^ cell/mL for 15 min at 37 °C, 30 min at RT or at 4 °C overnight. Cell cycles were analyzed with flow cytometry (BD Fortessa).

*Statistical analysis:* Data represent the mean ± SD from at least three independent determinations. The significance of differences between more than three samples was analyzed by one-way ANOVA and post hoc test, whereas the significance between two samples was analyzed by the Mann–Whitney U test using the GraphPad Prism software ver. 6.0 (GraphPad Software, San Diego, CA, USA), and a *p*-value less than 0.01 was considered statistically significant.

## 3. Results

### 3.1. Formulation and Characterization of PTX-Nanoparticles

In order to achieve a sustained and prolonged effect of PTX after its administration, it was encapsulated inside polymeric particles, composed of proprietary P polymer [16,17], The physicochemical characterization of these particles showed sizes of around 148 nm, with negative surface charges and 1.4% drug loading (Table 1 summarizes the results), indicating that, they were appropriate for parenteral administration, as already expected. In fact, these nanoparticles had already been prepared but included a targeting peptide to cross the blood–brain barrier, to treat glioblastoma cells [16,17,29].

### 3.2. Setting Up the pBAE Formulation for the Efficient Encapsulation of siRNAs

In our group, we have extensive expertise using our proprietary oligopeptide end-modified pBAE nanoparticles for the encapsulation of different kinds of nucleic acids [22,23,24,30]. In this work, we aimed to encapsulate the siRNA anti-survivin using the C32 polymer family (the most hydrophilic one), for their use in bladder cancer treatment. We selected five different oligopeptides combinations that worked efficiently for previous applications [23], and we started assessing the required N/P ratio to encapsulate a scrambled siRNA corresponding to the two anti-survivin siRNAs designed (see details in the Methods section). As shown in Figure 1, 10/1 and 25/1 ratios were not sufficient for the encapsulation of the siRNA, at least a 50/1 ratio was required to achieve an encapsulation higher than 60% of siRNA. For further experiments, 100/1 and 150/1 ratios were selected as a compromise between an efficient encapsulation with the lowest amount of polymer needed, being the siRNA, the active compound, the one to be maximized in the formulation.

### 3.3. Design of Experiments (DOE) for the Selection of pBAE Types

After the first EMSA analysis of encapsulation efficiency, the number of possible combinations was still high, since none of the conditions could be discarded. For this reason, a design of experiments (DoE) [31,32] was performed with the objective of rationally establishing the conditions for the most efficient nanoparticle synthesis. It was required to reduce the variables studied, so R and RK polymer combinations were selected for further experiments. In the next table (Table 2), the summary of the levels of the factors selected for the DoE is presented.

These factors were combined resulting in eight experiments (see Appendix A). The outputs selected for the analysis were: (1) nanoparticle size; (2) PDI and (3) storage stability in time. As shown in Figure 2, for nanoparticle size, the contribution of the polymer type was key, while it was not for PDI and time stability. Incubation time was very important to maintain the PDI and the stability over time, outputs in which the N/P ratio and the siRNA concentration also contributed.

Once the most influencing parameters were identified and after discarding the non-influencing factors, the experiments in with the expected size and PDI (size < 300 nm, PDI < 0.5) and with the smallest variation before and after incubation, were further studied in terms of size and PDI, as well as their stability over 60 min after preparation to finally select the level of each factor (Appendix A). These results are plotted in Figure 3. As clearly shown, experience 1 is the one showing the smallest size that was maintained after incubation time, so the levels of the factors that correspond to this experience were set up as the most appropriate ones: 0.03 mg/mL siRNA concentration, RK polymer combination, 25 mM buffer, preparation at 25 °C, 30 min incubation and 100/1 N/P ratio.

### 3.4. Modifying the Surface of the Particles to Enhance Their Stability

Although in the DoE the most stable formulation was chosen, clearly one-hour stability would not be enough for the clinical application of these particles. Consequently, and based on our previous studies, we selected the protease bromelain (PB), to coat nanoparticles and provide them with higher stability. Interestingly, it was described that this protein has the capacity of crossing mucosal barriers, required for the envisaged local intravesical delivery [33]. As shown in Figure 4, we were able to coat the particles without significantly modifying their characteristics in most conditions. In addition, the stability of nanoparticles over 2 h was maintained when we added the highest concentration of PB, and neither the size, the PDI, or the surface charge varied significantly. For these reasons, all concentrations could have been chosen and, consequently, we decided to use the highest one (0.33 mg/mL), as the PB concentration for the final formulation.

### 3.5. In Vitro Assays of pBAE-NPs Uptake and Transfection Efficiency

In vitro assays were performed with two bladder cell lines, selected to cover the different subtypes of bladder cancer, as usually performed in other antitumor studies [34], one being a monolayer homogeneously distributed adherent cells (T24), and the other forming clusters simulating adherent tumorspheres (RT4; see Appendix A); while RT4 represents adherent cells from a cell papilloma, T24 are adherent cells from transitional cell carcinoma.

First, the capacity of selected PB-coated RK pBAE nanoparticles to penetrate cells was qualitatively assessed by fluorescent microscopy, using a non-coding fluorescently labeled siRNA (F-siRNA). As shown in Figure 5, PB-coated, RK pBAE nanoparticles achieved high penetration in both cell lines, especially in RT4. These were surprising results since, growing in these clusters, RT4 cells were expected to be more difficult to penetrate due to the tight junctions between cells.

Further, the same nanoparticle type was prepared but encapsulating pGFP as a reporter plasmid, with the aim to study nanoparticle transfection efficiency. As shown in Figure 6, both cell lines were efficiently transfected by nanoparticles. Nevertheless, in this experiment, the trend shown in the uptake study was not observed. Here, the transfection levels were higher using the positive control, but it must be taken into account that Lipofectamine, although can efficiently transfect in vitro cultures, cannot be used in vivo due to toxicity issues. Comparing both cell lines, transfection was higher in T24, which could mean that although it seems that particles penetrate more in RT4 cells, the penetration may be superficial, being nanoparticles accumulated only to the border cells, not allowing the transfection of the inner ones.

### 3.6. In Vitro Antitumor Efficacy of PTX-NPs Monotherapy

Next, we tested, by means of colorimetric metabolic assays, the functionality of the engineered nanoparticles on bladder cell models. First, we checked the efficacy of PTX-NPs, in comparison to free PTX. As shown in Figure 7A, free PTX IC50 is strongly dependent on the cell line. As expected, the viability of T24 cells decreased as the concentration of PTX was increased to 600 nM. From this concentration, the loss of viability was kept at 20%. Thus, T24 cells were very sensitive to PTX, with an IC50 around 25 nM. For RT4 cells, the IC50 was much higher, around 300 nM. The decrease in viability was not as abrupt as seen in T24 cells and the viability observed was higher, from 45% to 60%. These cells showed significant resistance to PTX. This remarkable difference of viability between both cell types could be explained due to their differential sensitivity against PTX. As widely known, PTX is a drug that attacks the mitosis phase, preventing cells to divide [16,17,35]. Hence, it will be more effective in highly replicating cells. Accordingly, T24 cells have a doubling time of 20 h, whereas RT4 cells duplicate every 40 h. This important delay in the replication time could explain why the antitumor effect of PTX in RT4 after 3 days was less evident than in T24 cells that replicated in half of the time.

With established IC50 values, PTX-NPs were directly tested in different concentration ranges for each cell line. In addition, for RT4 cells, NPs were incubated for longer times to try to increase the effect. As shown in Figure 7B,C, the encapsulation of PTX inside the particles did not hamper its capacity to kill tumor cells. Nevertheless, while it significantly increased the mortality in RT4 cells, for T24 the effect was the opposite: the encapsulation of PTX resulted in lower mortality.

The higher viability of T24 cells treated with encapsulated PTX, as compared to naked PTX, was expected and attributed to the sustained release of PTX from P nanoparticles, as previously described [16]. When T24 cells were treated with 25 nM of encapsulated PTX, they had a 64% of cell viability, in comparison to the 53% achieved with the same concentration of free PTX. Accordingly, it was decided that PTX-NPs will be used at 25 nM, in combination with siRNA-NPs, in order to be able to detect a potential synergistic effect. Regarding RT4 cells, we decided to study the cytotoxic effect of PTX after 6 days, because of their slow replication rate and the high viabilities achieved in the IC50 assay after 3 days of treatment (Figure 7A). As shown in Figure 7C, encapsulated PTX induced a higher cytotoxic effect than free PTX, which can be again attributed to the sustained drug release. As we previously published [16], it takes some days to release PTX from the NPs, so we hypothesize here that the accessible concentration of PTX to cells after 6 days treatment of the initial single dose is higher for encapsulated PTX than for naked drug.

### 3.7. In Vitro Antitumor Efficacy of pBAE-NPs Monotherapy

Next, we studied the antitumor efficacy of pBAE-NPs encapsulating two different anti-survivin siRNAs (see structure in Appendix A), selected from a bibliographic search [20,21,36,37]. Before assessing the capacity of the particles to produce tumor cell death, we confirmed that the siRNA downregulated the survivin gene in these specific tumor cell lines by Western blot. As shown in Figure 8A and quantified in Figure 8B, transfection of cells with any of the two siRNAs against survivin induced a decrease in protein expression when compared to the negative control (untreated cells), especially for siRNA-1 in T24 cells, as expected from previous bibliography [38]. For the RT4 cell line, both siRNA tested achieved a similar silencing efficiency, although not as high as the one obtained for T24 cells (Figure 8B).

Next, the antitumor capacity was studied over time, for T24 cells, since it was the cell line showing the highest silencing effect (Figure 9A). As expected from silencing experiments, siRNA-1 achieved a higher killing effect, showing the highest decrease in cell viability (around 50%) after 3 days. This could be explained because, after long periods of time, cells that have not been killed are able to replicate again, hence increasing the cell culture viability. This could be also attributed to the degradation of the siRNA after the 3-day incubation time, which means that repeated doses could be required for the clinical use of the expected therapy. These data were in agreement with the previous bibliography. Yang et al., for example, showed a 30% decrease in cell viability and proliferation of T24 bladder cancer cells in vitro for 3 days when using the same survivin siRNA-1 oligonucleotide [35]. Interestingly, the efficacy of the present treatment was higher thanks to the use of pBAE NPs with the PB coating. In addition, other reports, such as the study of Grdina et al., needed a much higher concentration (100 nM vs. 16 pM used in the present work) to obtain similar cell killing efficiencies in models of colon carcinoma [34]. Taking into account these results, we studied the viability of RT4 cells only at the 3-day time point (Figure 9B). Again, siRNA-1 achieved a more potent effect on killing tumor cells than siRNA-2. It is worth noting that, comparing the effect on both cell lines, the antitumor effects observed in T24 cells were higher than those for RT4 cells (40% vs. 60% survival). This was expected according to the silencing results previously observed (Figure 8). Consequently, in the following studies, only siRNA-1 was used.

### 3.8. In Vitro Efficacy of the Dual Therapy

Once the efficacy of both monotherapies was confirmed, we aimed to design a dual combination therapy, since we hypothesized a synergistic effect could take place. After 3 days of treatment with 25 nM PTX-NPs and/or 16 pM siRNA-1 pBAE-NPs, the viability of T24 cells treated with the combination treatment was around 25% whereas cells only treated with PTX or only transfected with the anti-Survivin siRNA-1 had a 60% and 45% of viability respectively. Thus, the combined treatment produced a statistically significant decrease in cell viability compared to both treatments administered separately (Figure 9C). For RT4 cells, the experiment setup was the same, with the exception of the PTX dose, which, was adjusted to 300 nM. It is worth remarking that the dose of PTX is different between cell lines because it was adjusted according to the results of the monotherapies above (see Figure 7). As clearly seen (Figure 9D), the trend to an increase in the effect when combining both therapies is clear, although the effect, as already seen for monotherapies, is not as potent as for T24 cells. Nevertheless, for both cell types, the viability decrease was observed even with the scrambled siRNAs used as a negative control. Consequently, the mortality could not be attributed to the selective silencing of survivin, but to an unspecific effect of pBAE-NPs. We hypothesized that this effect could be explained due to the fact that the addition of PTX right after the incubation with PBAE-NPs could prevent cells from recovering from the transient toxicity associated with pBAEs transfection (produced due to a very high local concentration). In order to check this hypothesis, PTX-NPs were added to cells 24 h after siRNA treatment and cell viability was assessed 3 days later. Although in this case, a very slight decrease in the viability was observed with anti-survivin siRNA-1 pBAE-NP, this decrease was not significant in any of the cell lines tested. Therefore, although we may be reducing the unspecific toxicity mentioned above, the results were not conclusive.

### 3.9. Influence of the Cell Cycle Arrest on Dual Therapy Efficacy

At this point, we wondered why both treatments seemed to work when used as monotherapies, but not when combined. Thus, we performed a deeper study of survivin expression. In previous studies, nuclear and cytoplasmatic survivin isoforms were reported to have an almost identical structure that could not be differentiated by siRNAs or by antibodies [36], although they had different functions [37]. While the nuclear isoform might control cell division and proliferation, the cytoplasmic presence of survivin may be associated with cell survival and apoptosis [39]. The expression of both isoforms is not equal in all bladder cancer cells. Different studies showed that the nuclear expression of survivin was present only in 60% of TCC studied [40,41]. Taking these data into account, the expression of survivin was assessed in T24 and RT4 cell lines by fluorescent microscopy. As shown in Figure 10A, survivin expression in T24 cells was spread in the whole cell volume, including nuclear localization, while RT4 survivin expression was found preferentially accumulated in the cytoplasm.

Therefore, we hypothesized that we were only blocking the expression of a nuclear isoform of the survivin. This would stop the cell cycle, as described previously [9] impairing the cytotoxic effect of PTX because it can only kill dividing cells. Accordingly, we performed an analysis of the cell cycle (see details in Appendix A and summary in Figure 10B,C) and we observed that the inhibition of survivin produced a different cell cycle effect depending on cell line. In fact, only in T24 cells, G2M stage was significantly increased after 2 and 3 days of treatment with anti-survivin siRNA-1. According to previous literature [42], survivin regulates the cell cycle, with overexpression in the G2M stage. This is because T24 survivin is mostly located in the cell nucleus. Altogether, these data explain why the combination of PTX and siRNA against survivin did not induce a specific synergistic effect in T24 cells.

In the case of RT4 cells, their cell cycle was not influenced by the treatment (Figure 10C), which can be explained by the cytoplasmatic localization of survivin in this cell line (Figure 10A). The decrease in viability observed previously (Figure 9), where RT4 cells showed the viability of 60% after anti-survivin siRNA-1 transfection could have been produced by the inhibition of cytoplasmic survivin, which induces apoptosis [21,43]. Previously, we studied the amount of survivin expressed by RT4 and T24 cells in a Western blot assay. As it is shown in Figure 8, the levels of survivin expression in RT4 cells were much higher than those of T24 cells. We hypothesize that probably this fact might be the reason why no synergistic effect was observed when PTX was combined with the siRNA treatment in RT4 cells. The silencing of survivin could be enough to produce an increase in cell apoptosis but not enough to induce a decrease in chemoresistance against PTX.

## 4. Discussion

Bladder cancer remains among the ten most common cancers worldwide and clinical guidelines have not improved notably in the last years [1,2]. For this reason, the need for innovative therapeutic strategies is still a medical need. In this context, we aimed to develop here a dual therapy consisting of a chemotherapeutic drug with a gene-targeted therapy.

The chemotherapeutic drug selected was paclitaxel, due to its extended use for bladder cancer, among others. However, a major problem in the long-term efficacy of paclitaxel and other chemotherapeutics is the development of drug resistance, related to worse survival rates. Many studies have indicated that chemoresistance is induced by the overexpression of a set of genes related to the apoptotic route. This is the main reason why the rationale for a combined therapy based on gene silencing stands to be important [44]. Among these genes, survivin is attracting great attraction as one of the most relevant. It is an inhibitor of apoptosis protein (IAP) involved in many cellular responses to stress, presented in different subcellular compartments. Survivin is hardly detected in healthy adult cells, while overexpressed in fetal and tumor tissue [10,18]. Its relationship with the development of a wide variety of cancers, such as colon carcinomas, breast cancer, retinoblastoma, sarcomas and leukemias, has been clearly proven [9,10,36,38]. Survivin overexpression is associated not only with chemoresistance but with radioresistance, tumor growth, migration and aggressiveness and unfavorable clinical outcomes, where DNA damage takes place, producing survivin expression to be increased, thus resulting in a decrease in apoptosis [9,18,20,38]. Consequently, many strategies to downregulate its expression appeared and several studies demonstrated that the downregulation of survivin mRNA is associated with decreased tumor growth and sensitization to radiation and chemotherapeutic agents [42,45].

One of the most relevant and efficient forms to downregulate genes is the use of small interfering RNA (siRNA), a type of short double-stranded RNA that can specifically down-regulate survivin expression [18,44]. Nevertheless, oligonucleotides such as siRNA still show an important bottleneck step: their stability in biological media is compromised by the presence of nucleases, so a physical barrier between them and the biological media is required [46,47].

Therefore, it is clear that combining survivin inhibitors with paclitaxel would be a promising alternative, improved when using a nanomedicine strategy. Here, we propose this combination through the controlled delivery of both monotherapies: paclitaxel drug + survivin gene therapy, encapsulated in proprietary polymeric nanoparticles to achieve a synergistic effect killing cancer cells. Polymeric nanoparticles are used as the required technology to control the delivery of the active principles as well as to cross biological membranes [20].

Firstly, PTX was encapsulated in P polymer (see structure in Appendix A) [16]. These nanoparticles were previously used in our group for the treatment of glioblastoma multiforme, in a study where, thanks to the addition of a targeting peptide in the polymer, the particles efficiently crossed the blood–brain barrier and achieved a reduction of tumor growth and increase in animal survival [16]. Here, since we aim for the intravesicular administration, the addition of the peptide is not required for this local route. This is extremely advantageous in terms of therapeutic costs. These modified nanoparticles were synthesized, and their characterization enabled them to confirm they were appropriate for the intended use (Table 1).

Secondly, we synthesized poly(beta aminoester) nanoparticles for the encapsulation of siRNAs (see structure in Appendix A). These are also proprietary polymers from our group, long studied for the encapsulation of nucleic acids by us [22,23,24,48] and others [49,50,51], due to their advantageous properties in terms of reduced toxicity, that enables the administration of higher doses and, consequently, enhanced efficacy in gene transfection. Although previous studies already used pBAE nanoparticles for the encapsulation of siRNAs [15,23,30,52,53], and some encapsulated survivin siRNAs [54], here, two novelties stand as important. On the one hand, the use of a design of experiments (Figure 2 and Figure 3) for the selection of the methodological conditions for the formation of the nanoparticles. As far as we know, this is the first time that a rational method for the selection of these parameters was used to set up a formulation based on pBAEs. This is advantageous in terms of time-saving and efficiency of design. On the other hand, the intravesical delivery, enabled by the composition of nanoparticles [27,55]. To achieve so, after a first study of setting up the composition of the particles (Figure 1, Figure 2, Figure 3 and Figure 4), we selected C32 pBAE backbone, including 50% arginines and 50% lysines as terminal oligopeptides, with a coating of the protein bromelain, which enables the crossing of mucosal barriers [27,55]. An important point to highlight is the high plasmatic membrane penetration in both cell lines tested, especially in RT4 cells that grow forming clusters that were described as highly restrictive to transfection (Figure 5).

When used as monotherapies, both treatments showed high efficacies as antitumor therapies, tested in two cellular models of bladder cancer, representative of the papillary carcinoma (RT4) and carcinoma in situ (T24) cancer subtypes. The expected effect of PTX was confirmed by these in vitro studies, to further assess the IC50 that was used for combination therapy. In the case of pBAE-NPs, when they encapsulated both anti-survivin siRNAs, a decrease in cell viability was also observed, especially with siRNA-1 (Figure 9). However, the levels of mortality achieved by survivin silencing were far from those achieved when using paclitaxel. Although one could think that it is not worthy to develop these kinds of gene therapies if they are not able to overpass the effects of traditional chemotherapies, it must be taken into account the high toxicity produced when these chemotherapies such as paclitaxel are delivered. Therefore, although their great efficacies in vitro, doses must be adjusted down to avoid these side effects. On the contrary, since siRNAs target directly survivin expression and it is only overexpressed in tumor genes, the activity will be more localized into tumors. Additionally, the encapsulation of both active ingredients in nanoparticles and the local delivery of the therapy will synergistically contribute to avoiding the side effects.

Unfortunately, although it seemed clear that the combined therapy would be better, the results were unexpected. The synergistic effect was not observed. As already commented in the Results section, this can be attributed to the subcellular mechanism of action of paclitaxel together with survivin isoforms. We found that T24 showed a preferential survivin nuclear localization, usually attributed to regulation of mitosis [18]; for RT4 it was more cytoplasmatic (Figure 10). This cytoplasmatic expression was associated with advanced-stage tumors with chemoresistance [9]. This localization played an unexpected role in paclitaxel action. The drug, killing only dividing cells, was not able to synergize with survivin silencing therapy in T24 cells, since the downregulation of survivin produced a cell cycle arrest, only in T24 cells (Figure 10). In RT4 cells, the non-synergistic effect of the dual therapy should have been produced, as it was found for other types of tumor cells with cytoplasmatic expression of survivin [9]. Consequently, another factor must play a role. The lack of synergistic effect could be attributed to the initial higher survivin expression in RT4 cells as compared to T24 and the low doses of survivin siRNAs used. Therefore, although paclitaxel killed the cells, survivin inhibition was not enough to achieve the expected synergistic effect. In future studies, higher doses of survivin siRNA could be tested, if they are not toxic, to check if the envisaged synergistic effect is observed, at least, in cells with a preferential expression of survivin in the cytoplasm. Accordingly, a modification of the dosage pattern established could also be envisaged. While here we administered first the silencing therapy to sensitize tumor cells, in previous reports, the inverse pattern was followed [10]. Another alternative that could be feasible for future studies would be the silencing of other proteins involved in the cell growth and apoptosis control, such as the mammalian target of rapamycin (mTOR) or *wnt* proteins, both inducers of survivin overexpression [18].

This is proof of the importance of specific cell mechanisms when setting up new combination therapies for cancer, currently a vastly extended practice. For this reason, we recommend performing the efficacy studies of any combination therapy using more than one cell line and together with basic sciences studies of subcellular mechanisms involved, since, as we demonstrated here, although using similar cell lines, the results of the same therapy could be very different.

## 5. Conclusions

In summary, we designed a dual antitumor therapy for bladder cancer. It is based on paclitaxel encapsulated in P-polymer nanoparticles and siRNA survivin-loaded poly(beta aminoester) nanoparticles. Our results suggest that both nanoparticles were efficiently formed, with properties that could enable their intravesical administration, thus reducing invasiveness and the not-selective intravenous route. Each monotherapy was able to significantly reduce the growth of two bladder tumor cells in vitro. Nevertheless, the combination therapy did not enhance the tumor cell killing. This unexpected effect was further studied and attributed to the presence of nuclear vs. cytoplasmatic survivin isoforms combined with the cell cycle arrest produced by nuclear survivin, which is incompatible with the paclitaxel mode of action. Therefore, we could not confirm our hypothesis: survivin silencing does not sensitize bladder cell lines to paclitaxel.

## Figures and Tables

**Figure 1 pharmaceutics-13-01959-f001:**
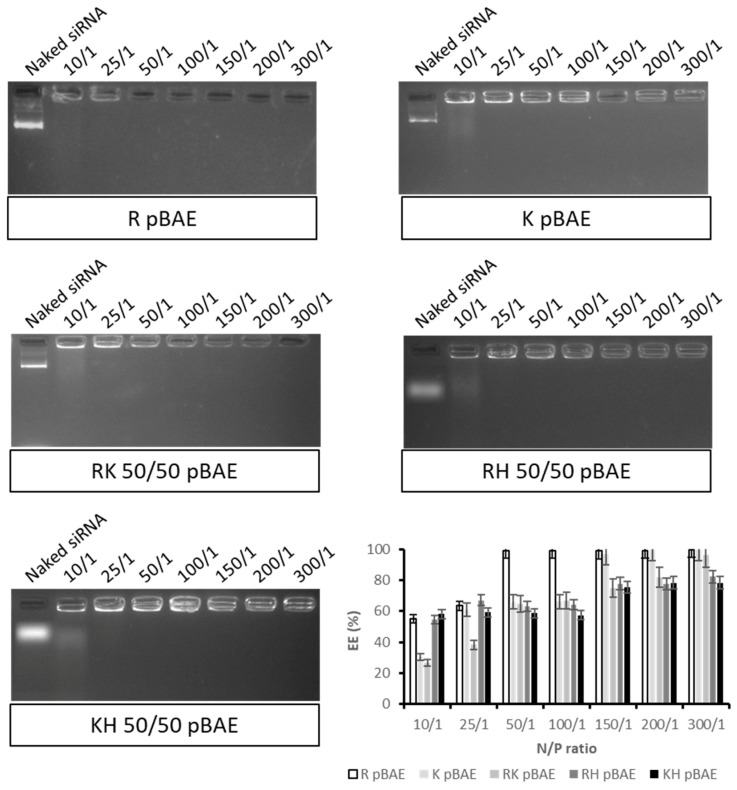
EMSA assay for the determination of siRNA encapsulation. Testing different N/P ratios and oligopeptide combinations, encapsulating a scrambled siRNA non-targeting pool.

**Figure 2 pharmaceutics-13-01959-f002:**
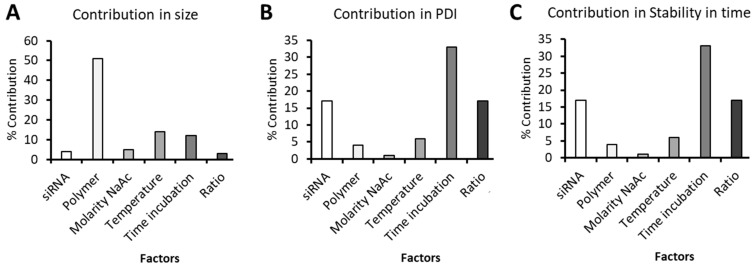
Results of the DoE. Contributions of the factors (in %) on: (**A**)—Size; (**B**)—PDI and (**C**)—Stability in time.

**Figure 3 pharmaceutics-13-01959-f003:**
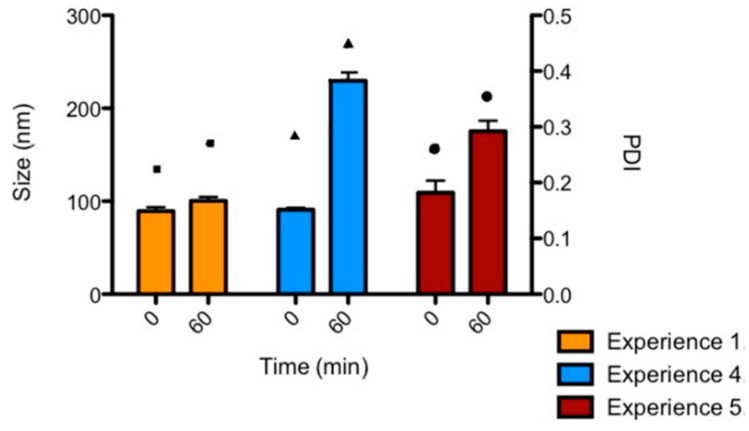
Results of the DoE. Contributions of the factors (in %) on: Size (bars, left axis) and PDI (squares, traangles and circles, right axis).

**Figure 4 pharmaceutics-13-01959-f004:**
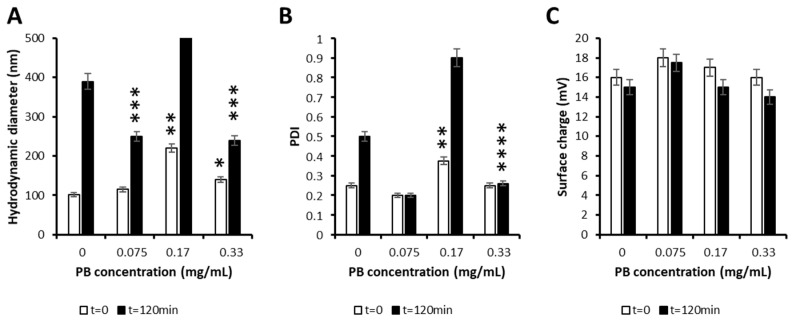
Physicochemical characterization of PB-coated pBAE-NPs. (**A**)—Size (nm); (**B**)—PDI; and (**C**)—Surface charge of PB-coated nanoparticles, as a function of PB concentrations, at initial and after 120 incubation. Statistical test comparing each condition with nanoparticles without the coating (at initial or final times). * *p* < 0.05; ** *p* < 0.01; *** *p* < 0.001; **** *p* < 0.0001.

**Figure 5 pharmaceutics-13-01959-f005:**
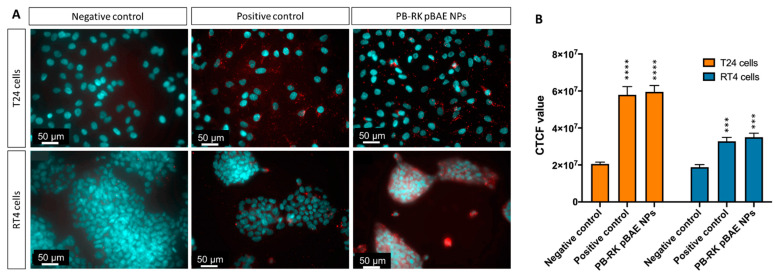
Nanoparticles’ uptake. (**A**)—Micrographic of T24 and RT4 cells, after being incubated with 0.03 mg/mL F-siRNA, using different transfecting agents. Nuclei were stained with DAPI. (**B**)—Relative quantification of the uptake, given as CTCF values. *** *p* < 0.001; **** *p* < 0.0001.

**Figure 6 pharmaceutics-13-01959-f006:**
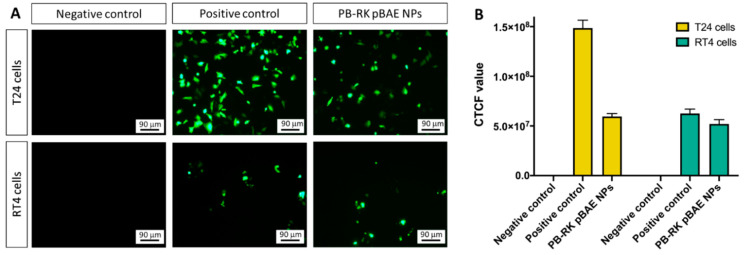
Nanoparticles transfection. (**A**)—Micrographies of T24 and RT4 cells, after being incubated with 0.03 mg/mL pGFP, using different transfecting agents. Positive control was performed with Lipofectamine 2000. (**B**)—Relative quantification of the transfection, given as CTCF values.

**Figure 7 pharmaceutics-13-01959-f007:**
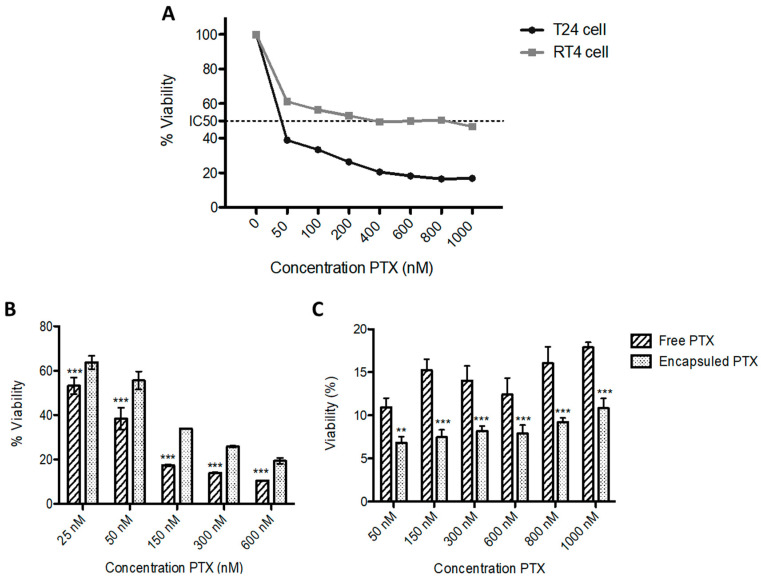
Antitumor efficacy of PTX-NPs. MTT results of cell viability, after being incubated, to increasing concentrations of: (**A**)—Naked PTX, for 3 days; (**B**,**C**)—naked PTX and PTX-NPs: (**B**)—T24 cells, 3 days incubation; (**C**)—RT4 cells, 6 days incubation. Dash line represents 50% viability. Statistics between equal doses of naked vs. encapsulated PTX. ** *p* < 0.01, *** *p* < 0.001.

**Figure 8 pharmaceutics-13-01959-f008:**
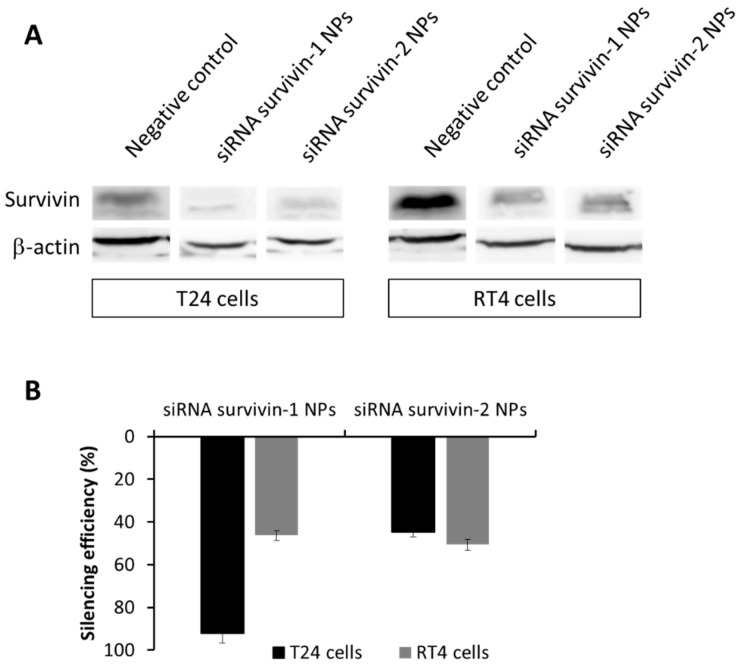
Survivin expression modulation by nanoparticles treatment. (**A**)—Western blot bands of T24 and RT4 survivin expression, after 3 days treatment; and (**B**)—Quantification of the silencing efficiency (%).

**Figure 9 pharmaceutics-13-01959-f009:**
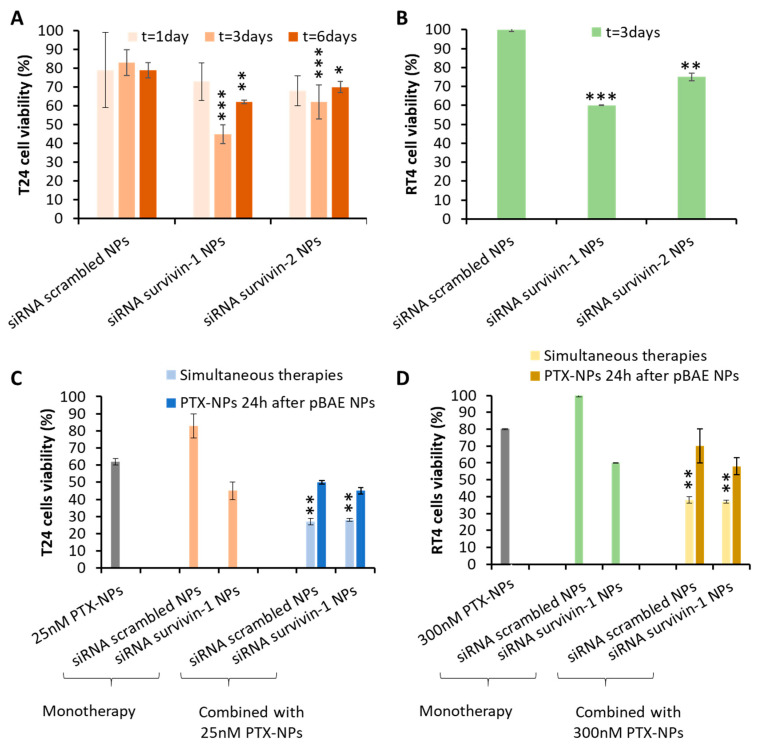
Tumor cells killing by pBAE nanoparticles treatment. Percentage of cell viability, after incubation with nanoparticles loaded with different siRNAs: (**A**,**B**)—pBAE-NPs monotherapies: (**A**)—T24 cells, at different times; and B—RT4 cells, at 3 days; and (**C**,**D**)—Combined therapies study: (**C**)—T24 cells; and (**D**)—RT4 cells. Statistical tests comparing with the effect of the scrambled siRNA (**A**,**B**) or with monotherapies (**C**,**D**). * *p* < 0.05; ** *p* < 0.01; *** *p* < 0.001.

**Figure 10 pharmaceutics-13-01959-f010:**
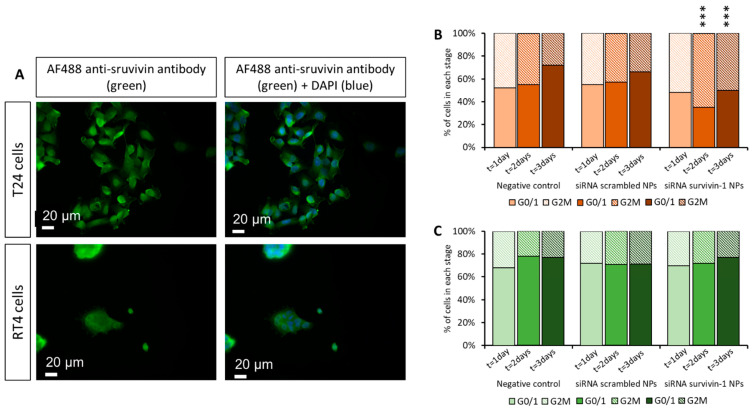
Intracellular and cell cycle studies. (**A**)—Fluorescence micrographs of subcellular localization of survivin isoforms as a function of cell type; (**B**,**C**)—Quantification of cell cycle stages of: (**B**)—T24 cells and (**C**)—RT4 cells, as a function of the treatment and time. Statistical analysis performed to compare results with the negative (non-treated) cells. *** *p* < 0.001.

**Table 1 pharmaceutics-13-01959-t001:** PTX-NPs physicochemical characterization. To determine the hydrodynamic diameter (nm), PDI, surface charge (mV) and drug loading. N = 10 replicates.

Hydrodynamic diameter (nm)	148 nm
PDI	<0.15
Zeta potential (mV)	−18 mV
Drug loading	1.4%

**Table 2 pharmaceutics-13-01959-t002:** DoE design for the selection of the most efficient nanoparticles synthesis.

Factors	Low Level	High Level
**Concentration siRNA**	0.01 mg/mL	0.03 mg/mL
**Type of polymer**	R	RK
**Molarity NaAc**	10 mM	25 mM
**Temperature**	25 °C (RT)	37 °C
**Time of incubation**	10 min	30 min
**Ratio**	100:1	150:1

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
