# Peer review of "Role of Survivin in Bladder Cancer: Issues to Be Overcome When Designing an Efficient Dual Nano-Therapy"

_pharmaceutics, 2021, doi:10.3390/pharmaceutics13111959_

Round 1

Reviewer 1 Report

In this manuscript, a dual nano-therapeutic platform combining the chemotherapy and gene therapy was employed to treat bladder cancer, which performed the effective therapeutic efficacy in vitro. However, several issues should be addressed.

  1. To further enhance the quality of the manuscript, some other strategies to treat bladder cancer should be introduced and discussed, such as the approaches reported on CHINESE CHEMICAL LETTERS, 2020, 31 (6), pp.1387-1391; CHINESE CHEMICAL LETTERS, 2020, 31 (5), pp.1159-1161.
  2. For dual therapy, it is better to optimize the does ratio of PTX-NPs vs. siRNA-NPs.
  3. It should calculate the combination indexes of PTX-NPs plus siRNA-NPs using the results of MTT assay to evaluate the synergistic efficacy
  4. The strategy in the manuscript is plan to overcome the drug resistance of bladder cancer cells, thus, it is better to choose the cell with property of PTX-resistance.
  5. I wander whether it would achieve a further enhanced anti-tumor efficacy when the PTX and siRNA were co-loaded in the same NPs.

Author Response

  1. To further enhance the quality of the manuscript, some other strategies to treat bladder cancer should be introduced and discussed, such as the approaches reported on CHINESE CHEMICAL LETTERS, 2020, 31 (6), pp.1387-1391; CHINESE CHEMICAL LETTERS, 2020, 31 (5), pp.1159-1161.

We acknowledge the reviewer for this information. Although the recommended articles deal with diagnosis of bladder cancer instead of treatment, following reviewer comments, in the revised version of our article, the references indicated have been added and the text of the introduction has been modified as it follows: “Bladder cancer is the 10th most diagnosed cancer with estimated 440,000 new cases worldwide on 2020, accounting for a 5-year prevalence up to 1.7M people[1], [2]. The most common histologic type of bladder cancer is urothelial carcinoma, which was formerly known as transitional cell carcinoma (TCC), due to the fact that the transitional cells lo-cated at the outsider layer of the bladder are the ones transformed into cancer cells[3]. Although current diagnosis methods are highly invasive, and bladder cancer remains complex and difficult to identify[4], [5], it is well known that Aall types of bladder cancer start in the inner lining of the bladder but from different cell types: urothelial cells (95% of bladder cancer), squamous cells (4%) and glandular mucus cells (1%). Approximately 75% of patients are non-muscle invasive TCC and have a 5-year survival rate between 88% and 98%[2], [6]. The other 25% of patients diagnosed with TCC are muscle invasive in stages between 1 and 4. Depending on the stage of the muscle invasive TCC the survival rate can range between 80% in the 5 years after diagnosis until 5% of survival with a cancer in stage 4[1].”

  1. For dual therapy, it is better to optimize the does ratio of PTX-NPs vs. siRNA-NPs.

We agree with the reviewer and we intended to do that. As already stated in the first version of the submitted manuscript, we adjusted the PTX dose as a function of cell line in order to take into account the effective doses found in monotherapies study. Nevertheless, to remark this issue, and according to the reviewer, we have modified the main text to clarify it. In the revised version of the manuscript, it reads as: “. It is worth remarking that the dose of PTX is different between cell lines because it was adjusted according to the results of the monotherapies above (see Figure 7).” In section 3.8. In addition, thanks to this comment, we realized that Figure 8 contained an error in the PTX concentration, and for this reason, it has been changed.

  1. It should calculate the combination indexes of PTX-NPs plus siRNA-NPs using the results of MTT assay to evaluate the synergistic efficacy

Although we agree with the reviewer that it would be interesting to add the combination index value, in here, since we only tested one concentration of the siRNA monotherapy, we cannot calculate it. Nevertheless, we added a comment on the main manuscript referring to that. In the revised version of the manuscript it reads: “. No combination index was calculated because we did not intend to test various concen-trations of the siRNA monotherapy. Thus, at the concentrations selected, the combined treatment produced a statistically significant decrease in cell viability compared to both treatments administered separately (Figure 9C).” in section 3.8.

  1. The strategy in the manuscript is plan to overcome the drug resistance of bladder cancer cells, thus, it is better to choose the cell with property of PTX-resistance.

We appreciate the reviewer comment. Nevertheless, our first aim was to try to enhance the efficacy of a combination therapy in a broad cohort of patients, without a bias for the PTX ones.

  1. I wander whether it would achieve a further enhanced anti-tumor efficacy when the PTX and siRNA were co-loaded in the same NPs.

We really appreciate this comment and it could be one of the strategies in the future.

Reviewer 2 Report

General comments
The authors in this manuscript provide the experimental data on a novel dual nano-treatment for bladder cancer to target survivin in cancer cells. It is 
based on paclitaxel encapsulated in P-polymer nanoparticles and siRNA survivin-loaded poly(beta aminoester) nanoparticles. 
Results clearly showed the inhibitory effect of each treatment but without synergism, highlighting that silencing of survivin does not sensitize cells to currently applied chemotherapies.

The manuscript to me is, in general, clearly written. The science and technical execution of the study is of good quality. The study is solid and the data, in general, support the conclusions. The theory, logic, and experimental design are easy to follow and in general, make sense.

Specific comments  

Title: change Surviving to Survivin.

Abbreviations should be properly used such as mQ H2O at line 119 and PDI at table one,...etc.

Line 223: add how you quantified Western blot data

Line 278: change stablish to establish.

In suppl Figures: change Fig. 3 and 4 to Figure S3 and S4.

Overall, I believe the improved version of the manuscript will be of interest to the field of bladder cancer. Therefore, it should be recommended for publication in pharmaceutics after minor revision. 

Author Response

The authors in this manuscript provide the experimental data on a novel dual nano-treatment for bladder cancer to target survivin in cancer cells. It is  based on paclitaxel encapsulated in P-polymer nanoparticles and siRNA survivin-loaded poly(beta aminoester) nanoparticles. 
Results clearly showed th
e inhibitory effect of each treatment but without synergism, highlighting that silencing of survivin does not sensitize cells to currently applied chemotherapies.

The manuscript to me is, in general, clearly written. The science and technical execution of the study is of good quality. The study is solid and the data, in general, support the conclusions. The theory, logic, and experimental design are easy to follow and in general, make sense.

Specific comments  

Title: change Surviving to Survivin.

We apologize for this spelling error. We have changed in the revised version of the manuscript.

Abbreviations should be properly used such as mQ H2O at line 119 and PDI at table one,...etc.

We have amended these errors, we acknowledge the reviewer comments.

Line 223: add how you quantified Western blot data

We have added this information on the experimental section. We apologize for this missing information. In line 225 of the revised version of the manuscript, it reads as follows: “). Images were quantified using ImageJ software, by normalizing the intensity of each band to the housekeeping gene band intensity used here (actin).”.

Line 278: change stablish to establish.

We amended this error.

In suppl Figures: change Fig. 3 and 4 to Figure S3 and S4.

We have changed these numbers.

Overall, I believe the improved version of the manuscript will be of interest to the field of bladder cancer. Therefore, it should be recommended for publication in pharmaceutics after minor revision.

Reviewer 3 Report

In the present study, the authors developed two different polymeric nanoparticles to encapsulate PTX and survivin siRNA independently. Then, they tested them as dual therapy.

The data could not support the hypothesis that the dual therapy was effective to treat bladder cancer cells. The authors showed a considered amount of data, but they were not convincing to support the hypothesis that the dual treatment has advantage. The following issues were found:

  1. The polymer nanoparticles encapsulating PTX (PTX- 110 NP) showed low drug loading (1.4%) and the stability with time and encapsulation efficiency were not presented. The Design of experiments (DOE) should be developed also for this nanoparticles.

  1. The authors results are contradictory in many points when they discuss about the T24 and RT4 cell line:

Line 329: pBAE nanoparticles achieved a high penetration in both cell lines, especially in RT4….BUT this cells grow in these clusters,  expecting to be more difficult to penetrate

Figure 7 B and C: “the results for T24 and RT4 are opposite considering the Free PTX and encapsulated PTX”. For T24 the viability reduced when the cells were treated with free PTX and for RT4 when treated with encapsulated PTX.

Line 379: “Regarding RT4 cells, we decided to study the cytotoxic effect of PTX after 6 days” and T24 cell line were treated for 3 days. Considering the intravesical instillation to treatment of bladder cancer in vivo system, this treatment shows low permeability of the drugs into urothelium and the poor retention of them due to periodic urine (max 2h of retention). If the author needed more than 3 days to see the effect in cell lines, in the overlay to 3D models and/or in vivo  models they will probably not find any effect.

  1. Dual therapy: Line 433: After 3 days of treatment with 25nM PTX-NPs and/or 16pM siRNA-1 pBAE-NPs. If all the NPs were administrated together (dual) there was (or suppose to) a competition for activity. It would be difficult to correlate the cell cycle with the impairing the cytotoxic effect of PTX. The figure 10 A did not show clear the fluorescence of subcellular localization of survivin isoforms as a function of cell type.

  1. The Western Blot results in Figure 8A are not aligned. It is comparable to use the same membrane to detect each protein. Also, the β-actin (housekeeping protein) showed different concentration in the pics for T24 cells.

  1. The results presented were not good enough to prove that a combined therapy based on gene silencing has advantage considering the others therapies already used in the clinic.

Author Response

In the present study, the authors developed two different polymeric nanoparticles to encapsulate PTX and survivin siRNA independently. Then, they tested them as dual therapy.

The data could not support the hypothesis that the dual therapy was effective to treat bladder cancer cells. The authors showed a considered amount of data, but they were not convincing to support the hypothesis that the dual treatment has advantage.

Yes, we completely agree with the reviewer conclusion and, in fact, as we vastly described in the discussion, we could not confirm our hypothesis, and this is why we performed the study of the cell cycle (results on Figure 10) which explained the reason why the dual treatment did not synergized.

The following issues were found:

  1. The polymer nanoparticles encapsulating PTX (PTX- 110 NP) showed low drug loading (1.4%) and the stability with time and encapsulation efficiency were not presented. The Design of experiments (DOE) should be developed also for this nanoparticles.

 As already commented in the manuscript, the whole study of these nanoparticles was performed in a previous study that we published in Drug Delivery in 2018; and this is why it is not included here, to avoid repetition of the same results (DOI: 10.1080/10717544.2018.1436099). nevertheless, to clarify this issue, we have added a clearer sentence indicating where to find this information. In the revised version of the manuscript, it reads as follows: “ In fact, these nanoparticles had already been prepared and fully described by Fornaguera et al., but including a targeting peptide to cross the blood-brain barrier, to treat glioblasto-ma cells [16], [17], [30]. Their full physico-chemical characterization can be found there.” In section 3.1

  1. The authors results are contradictory in many points when they discuss about the T24 and RT4 cell line:

Line 329: pBAE nanoparticles achieved a high penetration in both cell lines, especially in RT4….BUT this cells grow in these clusters,  expecting to be more difficult to penetrate

We apologize for this misunderstanding. In order to clarify this point, we have changed the sentence. It reads as follows in the revised version of the manuscript: “As shown in Figure 5, PB-coated, RK pBAE nanoparticles achieved a high penetration in both cell lines, especially in RT4, where it was not expected. These were surprising results since, growing in these clusters, RT4 cells were hypothesized to be more difficult to penetrate due to the tight junctions between cells. Nevertheless, the results were better as expected, confirming that the formation of tumorespheres is not an impediment for pBAE nanoparticles cell penetration.”

Figure 7 B and C: “the results for T24 and RT4 are opposite considering the Free PTX and encapsulated PTX”. For T24 the viability reduced when the cells were treated with free PTX and for RT4 when treated with encapsulated PTX.

We do not see the controversy in here. We agree with the reviewer that the results are contradictory between both cell lines and this is exactly what we are explaining here. In fact, in the next paragraph of the text, a reasonable explanation is given to this issue.

Line 379: “Regarding RT4 cells, we decided to study the cytotoxic effect of PTX after 6 days” and T24 cell line were treated for 3 days. Considering the intravesical instillation to treatment of bladder cancer in vivo system, this treatment shows low permeability of the drugs into urothelium and the poor retention of them due to periodic urine (max 2h of retention). If the author needed more than 3 days to see the effect in cell lines, in the overlay to 3D models and/or in vivo  models they will probably not find any effect.

We acknowledge the reviewer comment and we will take into account for future work. In fact, what is needed is the retention of the particles during this time in the bladder epithelia. Although they are confidential, we have some results that indicate that our nanoparticles can strongly attach to epithelial cells, which could be used for their retention in the bladder environment.

  1. Dual therapy: Line 433: After 3 days of treatment with 25nM PTX-NPs and/or 16pM siRNA-1 pBAE-NPs. If all the NPs were administrated together (dual) there was (or suppose to) a competition for activity. It would be difficult to correlate the cell cycle with the impairing the cytotoxic effect of PTX. The figure 10 A did not show clear the fluorescence of subcellular localization of survivin isoforms as a function of cell type.

 We do not expect a competition for activity, because the mechanism of each monotherapy is completely different. While pBAE nanoparticles penetrate cells through endocytosis and interfere with the specific expression of the survivin mRNA, PTX nanoparticles do not penetrate cells by themselves and PTX acts at nuclear level, directly with the cell replication and not in the translation stage.

We agree in the difficulty of assigning the cell cycle arrest to any or other monotherapies when used in combination, but to discard their individual effect, we added a combination using scrambled mRNA, which does not produce any effects and thus, the observed effects are attributed to paclitaxel.

  1. The Western Blot results in Figure 8A are not aligned. It is comparable to use the same membrane to detect each protein. Also, the β-actin (housekeeping protein) showed different concentration in the pics for T24 cells.

We agree completely with the reviewer comment regarding western blots and to show the complete results, we have added the entire western membranes used in this study for editors’ evaluation. The reason they are not in the same membranes is for organization issues. Nevertheless, this is the reason why actin is added as a housekeeping gene, used for the normalization of the quantified signals. To clarify this point, we have added in the experimental part a sentence indicating how we performed this quantification.

  1. The results presented were not good enough to prove that a combined therapy based on gene silencing has advantage considering the others therapies already used in the clinic.

Again, we agree with the reviewer and, as clearly stated in the discussion and conclusions of our article, although this study does not support the use of a combined therapy, it is a good work to avoid repetition of the same combination therapies, longly hypothesized to be the solution for many incurable cancers. To be honest, although we would have preferred good synergistic results, we finally found that our hypothesis was not confirmed and found a reason for that: the cell cycle arrest.

Round 2

Reviewer 3 Report

Overall, the authors could explain the questions that I had. So, I believe that improved version of the manuscript will be interesting for bladder cancer field